# LCA: Loss Change Allocation for Neural Network Training

**Janice Lan**
Uber AI
janlan@uber.com

**Rosanne Liu**
Uber AI
rosanne@uber.com

**Hattie Zhou**
Uber
hattie@uber.com

**Jason Yosinski**
Uber AI
yosinski@uber.com

## Abstract

Neural networks enjoy widespread use, but many aspects of their training, representation, and operation are poorly understood. In particular, our view into the training process is limited, with a single scalar loss being the most common viewport into this high-dimensional, dynamic process. We propose a new window into training called Loss Change Allocation (LCA), in which credit for changes to the network loss is conservatively partitioned to the parameters. This measurement is accomplished by decomposing the components of an approximate path integral along the training trajectory using a Runge-Kutta integrator. This rich view shows which parameters are responsible for decreasing or increasing the loss during training, or which parameters "help" or "hurt" the network's learning, respectively. LCA may be summed over training iterations and/or over neurons, channels, or layers for increasingly coarse views. This new measurement device produces several insights into training. (1) We find that barely over 50% of parameters help during any given iteration. (2) Some entire layers hurt overall, moving on average against the training gradient, a phenomenon we hypothesize may be due to phase lag in an oscillatory training process. (3) Finally, increments in learning proceed in a synchronized manner across layers, often peaking on identical iterations.

## 1 Introduction

In the common stochastic gradient descent (SGD) training setup, a parameterized model is iteratively updated using gradients computed from mini-batches of data chosen from some training set. Unfortunately, our view into the high-dimensional, dynamic training process is often limited to watching a scalar loss quantity decrease over time. There has been much research attempting to understand neural network training, with some work studying geometric properties of the objective function [7, 20, 28, 24, 21], properties of whole networks and individual layers at convergence [4, 7, 15, 35], and neural network training from an optimization perspective [30, 4, 5, 3, 19]. This body of work in aggregate provides rich insight into the loss landscape arising from typical combinations of neural network architectures and datasets. Literature on the dynamics of the training process itself is more sparse, but a few salient works examine the learning phase through the diagonal of the Hessian, mutual information between input and output, and other measures [1, 25, 14].

In this paper we propose a simple approach to inspecting training in progress by decomposing changes in the overall network loss into a per-parameter *Loss Change Allocation* or *LCA*. The procedure for computing LCA is straightforward, but to our knowledge it has not previously been employed for investigating network training. We begin by defining this measure in more detail, and then apply it to reveal several interesting properties of neural network training. Our contributions are as follows:

1. We define the Loss Change Allocation as a per-parameter, per-iteration decomposition of changes to the overall network loss (Section 2). Exploring network training with this measurement tool uncovers the following insights.

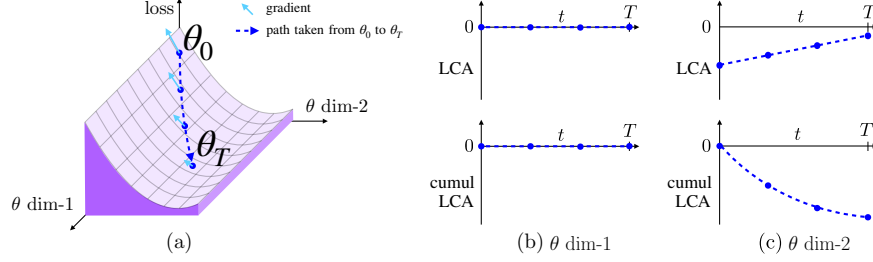

Figure 1: **(a)** Illustration of this paper's method on a toy two-dimensional loss surface. We allocate credit for changes to the model's training loss to individual parameters **(b)** $\theta$ dim-1 and **(c)** $\theta$ dim-2 by multiplying parameter motion with the corresponding individual component of the gradient of the training set. This partitions changes to the loss into individual *Loss Change Allocation (LCA)* components allows us to measure which parameters learn at each timestep, providing a rich view into the training process. In the example depicted, although both parameters move, the second parameter captures all the credit, as only its component of the gradient is non-zero.

2. Learning is very noisy, with only slightly over half of parameters helping to reduce loss on any given iteration (Section 3).

3. Some *entire layers* consistently drift in the wrong direction during training, on average moving *against* the gradient. We propose and test an explanation that these layers are slightly out of phase, lagging behind other layers during training (Section 4).

4. We contribute new evidence to suggest that the learning progress is, on a microscopic level, *synchronized* across layers, with small peaks of learning often occurring at the same iteration for all layers (Section 5).

## 2 The Loss Change Allocation approach

We begin by defining the Loss Change Allocation approach in more detail. Consider a parameterized training scenario where a model starts at parameter value $\theta_0$ and ends at parameter value $\theta_T$ after training. The training process entails traversing some path $P$ along the surface of a loss landscape from $\theta_0$ to $\theta_T$. There are several loss landscapes one might consider; in this paper we analyze the *training* process, so we measure motion along the loss with respect to the *entire training set*, here denoted simply $L(\theta)$. We analyze the loss landscape of the training set instead of the validation set because we aim to measure training, not training confounded with issues of memorization vs. generalization (though the latter certainly should be the topic of future studies).

The approach in this paper derives from a straightforward application of the fundamental theorem of calculus to a path integral along the loss landscape:

$$L(\theta_T) - L(\theta_0) = \int_C \langle \nabla_\theta L(\theta), d\theta \rangle \tag{1}$$

where $C$ is any path from $\theta_0$ to $\theta_T$ and $\langle \cdot, \cdot \rangle$ is the dot product. This equation states that the change in loss from $\theta_0$ to $\theta_T$ may be calculated by integrating the dot product of the loss gradient and parameter motion along a path from $\theta_0$ to $\theta_T$. Because $\nabla_\theta L(\theta)$ is the gradient of a function and thus is a conservative field, any path from $\theta_0$ to $\theta_T$ may be used; in this paper we consider the path taken by the optimizer during the course of training. We may approximate this path integral from $\theta_0$ to $\theta_T$ by using a series of first order Taylor approximations along the training path. If we index training steps by $t \in [0, 1, ..., T]$, the first order approximation for the change in loss during one step of training is the following, rewritten as a sum of its individual components:

$$L(\theta_{t+1}) - L(\theta_t) \approx \langle \nabla_\theta L(\theta_t),\ \theta_{t+1} - \theta_t \rangle \tag{2}$$

$$= \sum_{i=0}^{K-1} (\nabla_\theta L(\theta_t))^{(i)} (\theta_{t+1}^{(i)} - \theta_t^{(i)}) := \sum_{i=0}^{K-1} A_{t,i} \tag{3}$$

where $\nabla_\theta L(\theta_t)$ represents the gradient of the loss of the whole training set w.r.t. $\theta$ evaluated at $\theta_t$, $v^{(i)}$ represents the $i$-th element of a vector $v$, and the parameter vector $\theta$ contains $K$ elements. Note that while we *evaluate model learning* by tracking progress along the training set loss landscape

$L(\theta)$, training itself is accomplished using stochastic gradient approaches in which noisy gradients from mini-batches of data drive parameter updates via some optimizer like SGD or Adam. As shown in Equation 3, the difference in loss produced by one training iteration $t$ may be decomposed into $K$ individual *Loss Change Allocation*, or *LCA*, components, denoted $A_{t,i}$. These $K$ components represent the LCA for a single iteration of training, and over the course of $T$ iterations of training we will collect a large $T \times K$ matrix of $A_{t,i}$ values.

The total loss over the course of training will often decrease, and the above decomposition allows us to allocate credit for loss decreases on a per-parameter, per-timestep level. Intuitively, when the optimizer increases the value of a parameter and its component of the gradient on the whole training set is negative, the parameter has a negative LCA and is "helping" or "learning". Positive LCA is "hurting" the learning process, which may result from several causes: a noisy mini-batch with the gradient of that step going the wrong way, momentum, or a step size that is too large for a curvy or rugged loss landscape as seen in [14, 32]. If the parameter has a non-zero gradient but does not move, it does not affect the loss. Similarly, if a parameter moves but has zero gradient, it does not affect the loss. The sum of the $K$ components is the overall change in loss at that iteration. Figure 1 depicts a toy example using two parameters. Throughout the paper we use "helping" to indicate negative LCA (a contribution to the reduction of total loss), and "hurting" for positive LCA.

An important property of this decomposition is that it is *grounded*: the sum of individual components equals the total change in loss, and each contribution has the same fundamental units as the loss overall (e.g. nats or bits in the case of cross-entropy). This is in contrast to approaches that measure quantities like parameter motion or approximate elements of the Fisher information (FI) [16, 1], which also produce per-parameter measurements but depend heavily on the parameterization chosen. For example, the FI metric is sensitive to scale (e.g. multiply one relu layer weights by 2 and next by 0.5: loss stays the same but FI of each layer changes and total FI changes). Further, LCA has the benefit of being *signed*, allowing us to make measurements and interpretations when training goes backwards (Sections 3 and 4).

Ideally, summing up the $K$ components should equal $L(\theta_{t+1}) - L(\theta_t)$. In practice, the first order Taylor approximation is often inaccurate due to the curvature of the loss landscape. We can improve on our LCA approximation from Equation 2 by replacing $\nabla_\theta L(\theta_t)$ with $\frac{1}{6}(\nabla_\theta L(\theta_t) + 4\nabla_\theta L(\frac{1}{2}\theta_t + \frac{1}{2}\theta_{t+1}) + \nabla_\theta L(\theta_{t+1}))$, with the $(1, 4, 1)$ coefficients coming from the fourth-order Runge–Kutta method (RK4) [23, 17] or equivalently from Simpson's rule [31]. Using a midpoint gradient doubles computation but shrinks accumulated error drastically, from first order to fourth order. If the error is still too large, we can halve the step size with composite Simpson's rule by calculating gradients at $\frac{3}{4}\theta_t + \frac{1}{4}\theta_{t+1}$ and $\frac{1}{4}\theta_t + \frac{3}{4}\theta_{t+1}$ as well. We halve the step size until the absolute error of change in loss per iteration is less than 0.001, and we ensure that the cumulative error at the end of training is less than 1%. First order and RK4 errors can be found in Table S1 in Supplementary Information. Note that the approach described may be applied to any parameterized model trained via gradient descent, but for the remainder of the paper we assume the case of neural network training.

## 2.1 Experiments

We employ the LCA approach to examine training on two tasks: MNIST and CIFAR-10, with architectures including a 3-layer fully connected (FC) network and LeNet [18] on MNIST, and AllCNN [29] and ResNet-20 [9] on CIFAR-10. Throughout this paper we refer to training runs as "dataset–network", e.g., **MNIST–FC**, **MNIST–LeNet**, **CIFAR–AllCNN**, **CIFAR–ResNet**, followed by further configuration details (such as the optimizer) when needed.

For each dataset–network configuration, we train with both SGD and Adam optimizers, and conduct multiple runs with identical hyperparameter settings. Momentum of 0.9 is used for all SGD runs, except for one set of "no-momentum" MNIST–FC experiments. Learning rates are manually chosen between 0.001 to 0.5. See Section S7 in Supplementary Information for more details on architectures and hyperparameters. We also make our code available at `https://github.com/uber-research/loss-change-allocation`. Note that we use standard network architectures to demonstrate use cases of our tool; we strive for simplicity and interpretability of results rather than state-of-the-art performance. Thus we do not incorporate techniques such as L2 regularization, data augmentation, and learning rate decay. Since our method requires calculating gradients of the loss over the entire training set, it is considerably slower than the regular training process, but remains tractable for small to medium models; see Section S8 for more details on computation.

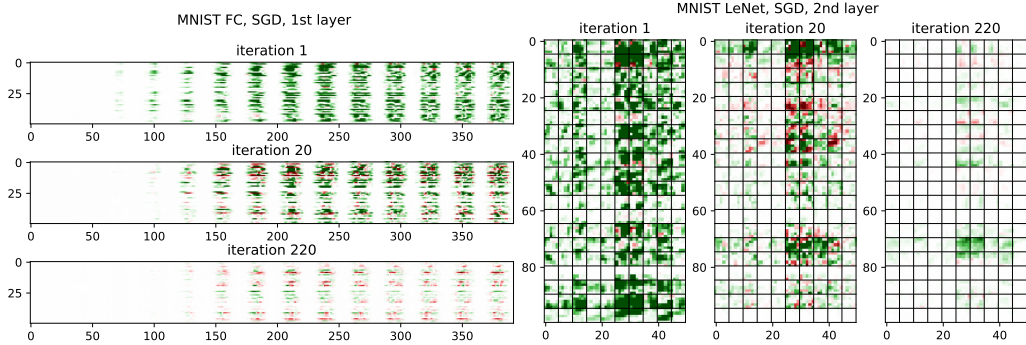

Figure 2: Frames from an animation of the learning process for two training runs. **(left)** The 1st layer of an MNIST–FC (full shape is $100{\times}784$, but only the upper left quarter is shown for better clarity). **(right)** The 2nd convolutional layer of an MNIST–LeNet (full shape is $40{\times}20$ of $5{\times}5$ blocks; only upper left quarter is shown). Each pixel represents one parameter. The LeNet layer shows $5{\times}5$ grids representing each filter, laid out by input channels (columns) and output channels (rows). Parameters that help (decrease the loss) at a given time are shown as shades of green. Parameters that hurt (increase the loss) are shown as shades of red. Larger magnitudes of LCA are darker and white indicates zero LCA. Iteration 20 is partly through the main drop in loss, and 220 is one full epoch. In MNIST–FC, we can see clusters spaced at intervals of 28 pixels, because these parameters connect to the flattened MNIST images. Learning is strongest in early iterations with mostly negative LCA, remains strong for many iterations but with more variance in LCA across parameters, and has greatly diminished by iteration 220, where much of learning is complete. The complete animations may be viewed at: `https://youtu.be/xcnoRnoVyXQ` and `https://youtu.be/EY3LoXmdkYU`.

## 2.2 Direct visualization

We calculate LCA for every parameter at every iteration and animate the LCA values through all the iterations in the whole training process. Figure 2 shows snapshots of frames from the video visualization. In such videos, we arrange parameters first by layer and then for each layer as two-dimensional matrices (1-D vectors for biases), and overlay LCA values as a heatmap. This animation enables a granular view of the training process.

We can also directly visualize each parameter versus time, granting each parameter its own training curve. We can optionally aggregate over neurons, channels, layers, etc. (see Section S2 for examples). A benefit of these visualizations is that they convey a large volume of data directly to the viewer, surfacing subtle patterns and bugs that can be further investigated. Observed patterns also suggest more quantitative metrics that surface traits of training. The rest of the paper is dedicated to such metrics and traits.

## 3 Learning is very noisy

Although it is a commonly held view that the inherent noise in SGD-based neural network training exists and is even considered beneficial [15], this noise is often loosely defined as a deviation in gradient estimation. While the minibatch gradient serves as as a suggested direction for parameter movement, it is still one step away from the actual impact on decreasing loss over the whole training set, which LCA represents precisely. By aggregating a population of per-parameter, per-iteration LCAs along different axes, we present numerical results that shed light into the noisy learning behavior. We find it surprising that on average *almost half of parameters are hurting* in every training iteration. Moreover, each parameter, including ones that help in total, *hurt almost half of the time*.

Table 1: Percentage of helping parameters (ignoring those with zero LCA) for various networks and optimizers, averaged across all iterations and 3 independent runs per per configuration.

|  | MNIST-FC, mom=0 | MNIST-FC | MNIST-LeNet | CIFAR-ResNet | CIFAR-AllCNN |
|---|---|---|---|---|---|
| SGD | $53.72 \pm 0.05$ | $57.79 \pm 0.16$ | $53.97 \pm 0.48$ | $50.66 \pm 0.14$ | $51.09 \pm 0.23$ |
| Adam | N/A | $55.82 \pm 0.09$ | $51.77 \pm 0.21$ | $50.30 \pm 0.004$ | $50.19 \pm 0.01$ |

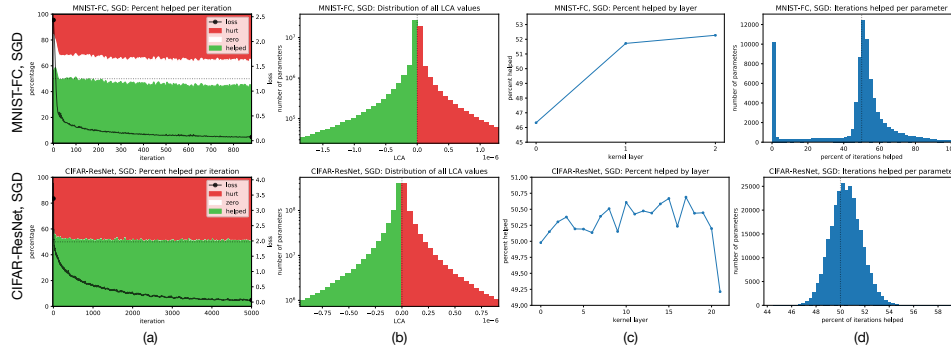

Figure 3: **(a)** Visualization of the percentage of parameters that helped, hurt, or had zero effect through training, overlaid with the loss curve of that run. **(b)** The distribution of helping and hurting LCA (zeros ignored) over the entire training, zoomed in to ignore 1% of tails. **(c)** Average percent of weights helping for each layer in network, curiously near 50% for all. **(d)** Histogram of the fraction of iterations each weight helped, showing that most weights swing back and forth between helping and hurting evenly. In every column the first row is MNIST–FC and second row CIFAR–ResNet, both trained with SGD. Notable facts: MNIST–FC shows a significant percent of weights with zero effect. Because MNIST has pixels that are never on, any first layer weights connected to those pixels cannot help or hurt. CIFAR–ResNet exhibits barely over 50% of parameters helping over the course of training, even during the period of significantly learning (loss reduction) from iteration 0 to 2000. Averaged over the entire run, only 50.66% of parameters helped (see Table 1). Note that in both runs we can see that in the earliest iterations, the percent of weights helping is higher, but only slightly.

**Barely over 50% of parameters help during training.** According to our definition, for each iteration of training, parameters can help, hurt, or not impact the overall loss. With that in mind, we count the number of parameters that help, hurt, or neither, across all training iterations and for various networks; two examples of networks are shown in Figure 3 (all other networks shown in Section S3). The data show that in a typical training iteration, close to half of parameters are helping and close to half are hurting! This ratio is slightly skewed towards helping in early iterations but stays fairly constant during training. Averaged across all iterations, the percentage of helping parameters for various network configurations is reported in Table 1. We see that it varies within a small range of 50% to 58%, with CIFAR networks even tighter at 50% to 51%. This observation also holds true when we look at each layer separately in a network; Figure 3(c) shows that all layers have similar ratios of helpful parameters.

**Parameters alternate helping.** Now that we can tell if each parameter is "helpful", "hurtful", or "neither"[1], we wonder if parameters predictably stay in the same category throughout training. In other words, is there a consistent elite group of parameters that always help? When we measure the percentage of helpful iterations per parameter throughout a training run, histograms in Figure 3(d) show that parameters help approximately half of the time, and therefore the training of a network is achieved by parameters alternating to make helpful contribution to the loss.

Additionally, we can measure the oscillations of individual parameters. Figure S7 shows a high number of oscillations in weight movement for CIFAR–ResNet on SGD: on average, weight movements change direction once every 6.7 iterations, and gradients change signs every 9.5 iterations. Section S3 includes these measures for all networks, as well as detailed views in Figure S8 suggesting that many of these oscillations happen around local minima. While oscillations have been previously observed for the overall network [32, 14], thanks to LCA, we're able to more precisely quantify the individual and net effects of these oscillations. As we'll see in Section 4, we can also use LCA to identify when a network is damaged not by oscillations themselves, but by their precise phase.

**Noise persists across various hyperparameters.** Changing the learning rate, momentum, or batch size (within reasonable ranges such that the network still trains well) only have a slight effect on the percent of parameters helping. See Section S3 for a set of experiments on CIFAR–ResNet with SGD, where percent helped always stays within 50.3% to 51.6% for reasonable hyperparameters.

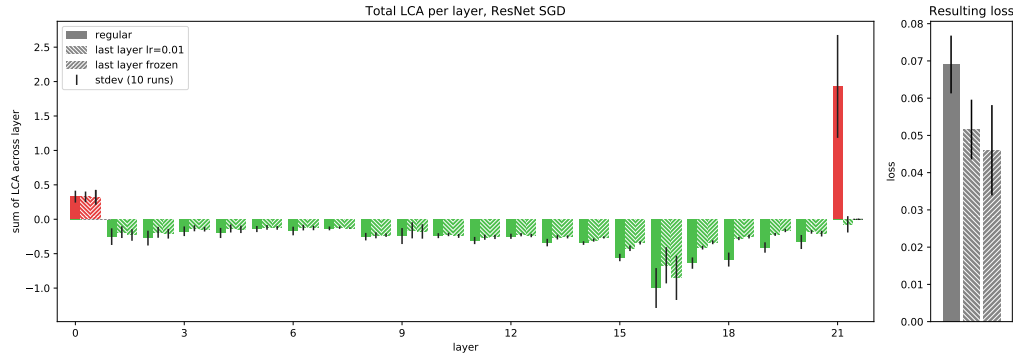

Figure 4: **(left)** LCA summed over all of training, for each layer, in CIFAR–ResNet trained with SGD. Bias and batch norm layers are combined into their corresponding kernel layers. Blue represents regular runs. Orange is with the last layer frozen at initialization. Note that the other layers, especially the adjacent few, do not help as much, but the difference in LCA of the last layer is greater than the total differences of the other layers helping less. Green is with the last layer at a 10x smaller learning rate than the rest of the network, showing similar layer LCAs as when the layer is frozen. **(right)** Resulting train loss and standard deviations for each run configuration. Means and standard deviations are over 10 runs for each experiment configuration.

**Learning is heavy-tailed.**   A reasonable mental model of the distribution of LCA might be a narrow Gaussian around the mean. However, we find that this is far from reality. Instead, the LCA of both helping and hurting parameters follow a heavy-tailed distribution, as seen in Figure 3(b). Figure S10 goes into more depth in this direction, showing that contributions from the tail are about three times larger than would be expected if learning were Gaussian distributed. More precisely, a better model of LCA would be the Weibull distribution with $k < 1$. The measurements suggest that the view of learning as a Wiener process [25] should be refined to reflect the heavy tails.

## 4   Some layers hurt overall

Although our method is used to study low-level, per-parameter LCA, we can also aggregate these over higher level breakdowns for different insights; individually there is a lot of noise, but on the whole, the network learns. The behavior of individual layers during learning has been of interest to many researchers [35, 22], so a simple and useful aggregation is to sum LCA over all parameters within each layer and sum over all time, measuring how much each layer contributes to total learning.

We see an expected pattern for MNIST–FC and MNIST–Lenet (all layers helping; Figure S11), but CIFAR–ResNet with SGD shows a surprising pattern: the *first and last layers consistently hurt training* (positive total LCA). Over ten runs, the first and last layer in ResNet hurt statistically significantly (p-values $< 10^{-4}$ for both), whereas all other layers consistently help (p-values $< 10^{-4}$ for all). Blue bars in Figure 4 shows this distinct effect. Such a surprising observation calls for further investigation. The following experiments shed light on why this might be happening.

**Freezing the first layer stops it from hurting but causes others to help less.**   We try various experiments freezing the first layer at its random initialization. Though we can prevent this layer from hurting, the overall performance is not any better because the other layers, especially the neighboring ones, start to help less; see Figure S13 for details. Nonetheless, this can be useful for reducing compute resources during training as you can freeze the first layer without impairing performance.

**Freezing the last layer results in significant improvement.**   In contrast to the first layer, freezing the last layer at its initialization (Figure 4) improves training performance (and test performance curiously; not shown), with p-values $< 0.001$ for both train loss and test loss, over 10 runs! We also observe other layers, especially neighboring ones, not helping as much, but this time the change in the last layer's LCA more than compensates. Decreasing the learning rate of the last layer by 10x (0.01 as opposed to 0.1 for other layers) results in similar behavior as freezing it. These experiments are consistent with findings in [12] and [8], which demonstrate that you can freeze the last layer in some networks without degrading performance. With LCA, we are now able to provide an explanation for when and why this phenomenon happens. The instability of the last layer at the start of training in [8] can also be measured by LCA, as the LCA of the last layer is typically high in the first few iterations.

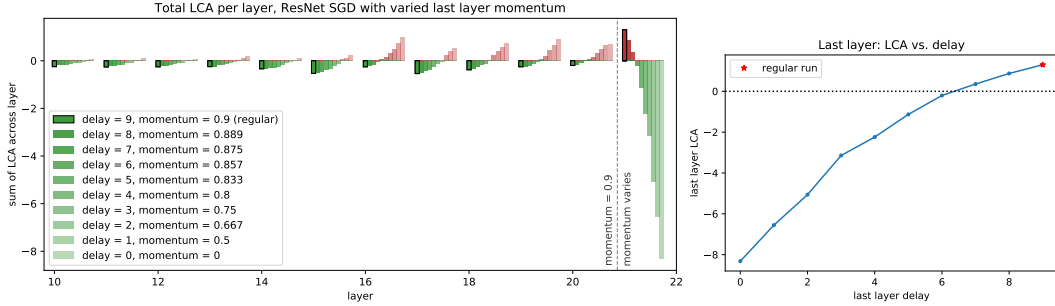

Figure 5: CIFAR–ResNet SGD with varying momentum for the last layer (and a fixed 0.9 for all other layers). Selected momentum values are derived from linear values of delay $[0, 1, 2, ..., 9]$ in a control system, where $\text{momentum} = \text{delay}/(\text{delay} + 1)$, and a delay of 9 corresponds to regular runs of 0.9 momentum. **(left)** LCA per layer (only the second half of the network is shown for better visibility; first half follows a similar trend, but less pronounced). As the last layer helps more, the other layers hurt more because they are relatively more delayed. **(right)** LCA of the last layer is fairly linear with respect to the delay.

**Phase shift hypothesis: is the last layer phase-lagged?** While it is interesting to see that decreasing the learning rate by 10x or to zero changes the last layer's behavior, this on its own does not explain why the layer would end up going *backwards*. The mini-batch gradient is an unbiased estimator of the whole training set gradient, so on average the dot product of the mini-batch gradient with the training set gradient is positive. Thus we must look beyond noise and learning rate for explanation. We hypothesize that the last layer may be *phase lagged* with respect to other layers during learning. Intuitively, it may be that while all layers are oscillating during learning, the last layer is always a bit behind. As each parameter swings back and forth across its valley, the shape of its valley is affected by the motion of all other parameters. If one parameter is frozen and all other parameters trained infinitesimally slowly, that parameters valley will tend to flatten out. This means if it had climbed a valley (hurting the loss), it will not be able to fully recover the LCA in the negative direction, as the steep region has been flattened. If the last layer reacts slower than others, its own valley walls may tend to be flattened before it can react.

A simple test for this hypothesis is as follows. We note that training with momentum 0.9 corresponds to an information lag of 9 steps (the mean of an exponential series with exponent .9)—each update applied uses information 9 steps old. To give the last layer an advantage, we train it with momentum corresponding to a delay of $n$ for $n \in \{9, 8, ..., 0\}$ while training all other layers as usual. As shown in Figure 5, this works, and the transition from hurting to helping (a lot) is almost linear with respect to delay! As we give the last layer an information freshness advantage, it begins to "steal progress" from other layers, eventually forcing the neighboring layers into positive LCA. These results suggest that it may be profitable to view training as a fundamentally oscillatory process upon which much research in phase-space representations and control system design may come to bear.

Beyond CIFAR–Resnet, other networks also show intriguingly heterogeneous layer behaviors. As we noted before, in the case of MNIST–FC and MNIST–LeNet trained with SGD, all layers help with varying quantities. An MNIST–ResNet (added specifically to see if the effect we see above is due to the data or the network) shows the last layer hurting as well. We also observe the last layer hurting for CIFAR–AllCNN with SGD (Figure S14) and multiple layers hurting for a couple of VGG-like networks (Figure S12). When using Adam instead of SGD, CIFAR–ResNet has a consistently hurting first layer and an inconsistently hurting last two layers. CIFAR–AllCNN trained with Adam does not have any hurting layers. We note that layers hurting is not a universal phenomenon that will be observed in all networks, but when it does occur, LCA can identify it and suggest potential candidates to freeze. Further, viewing training through the lens of information delay seems valid, which suggests that per-layer optimization adjustments may be beneficial.

## 5 Learning is synchronized across layers

We learned that layers tend to have their own distinct, consistent behaviors regarding hurting or helping from per-layer LCA summed across all iterations. In this section we further examine the per-layer LCA *during* training, equivalent to studying individual "loss curves" for each layer, and

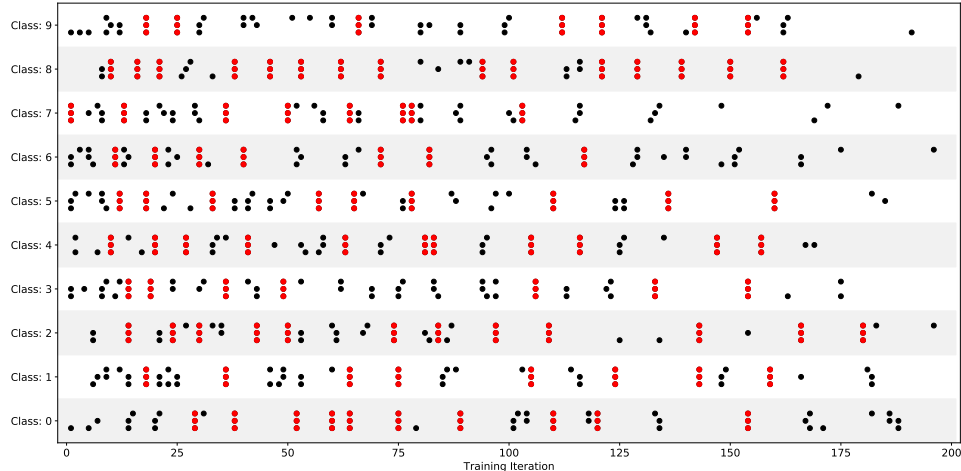

Figure 6: Peak learning iterations by layer by class on MNIST–FC. The same LCA data as in Figure S17 but seperated by class. We plot the top 20 iterations by LCA for each class and each layer, where that iteration represents a local minimum for LCA. The layers are ordered from bottom to top. Points highlighted in red represent iterations where all three layers had peak learning for that particular class. To measure the statistical significance of these vertical line structures in red, we simulate a baseline by shifting each layer in each class randomly by -2, -1, 0, or 1, 2 iteration. We find that the average number of vertical lines is 0.4 in the baseline and 9.4 in the actual network, and this difference is significant with a p-value < 0.001.

discover that the exact moments where learning peaks are curiously synchronized across layers. And such synchronization is not driven by only gradients or parameter motion, but both.

We define "moments of learning" as temporal spikes in an instantaneous LCA curve, local minima where loss decreased more on that iteration than on the iteration before or after, and show the top 20 such moments (highest magnitude of LCA) for each layer in Figure S17. We further decompose this metric by class (10 for both MNIST and CIFAR), where the same moments of learning are identified on per-class, per-layer LCAs, shown in Figure 6. Whenever learning is synchronized across layers (dots that are vertically aligned) they are marked in red. Additional figures on CIFAR–ResNet can be seen in Section S5. The large proportion of red aligned stacks suggests that learning is very locally synchronized across layers.

To gauge statistical significance, we compare the number of synchronized moments in these networks to a simple baseline: the number we would observe if each layer had been randomly shifted to one or two iterations earlier or later. We find that the number synchronized moments is significantly more than that of such a baseline (p-value $< 1^{-6}$). See details on this experiment in Section S5. Thus, we conclude that for these networks we've measured, learning happens curiously synchronously across layers throughout the network. We might find different behavior in other architectures such as transformer models or recurrent neural nets, which could be of interest for future work.

But what drives such synchronization? Since learning is defined as the product of parameter motion and parameter gradient, we further examine whether one of them is synchronized in the first place. By plotting in the same fashion of identified local peaks, we observe the synchronization pattern in gradients per layer is clearly different from that in LCA, either in terms of the total loss (Figure S18) or per-class loss (Figure S16). Since parameter motion (Figure S19) is the same across all classes, it alone doesn't drive the per-class LCA. We therefore conclude that the synchronization of learning, demonstrated by synchronized behavior in LCA movement (Figure 6), is strong, and comes from both parameter motion and gradient.

# 6   Conclusion

The Loss Change Allocation method acts as a microscope into the training process, allowing us to examine the inner workings of training with much more fine-grained analysis. When applied to various tasks, networks and training runs, we observe many interesting patterns in neural network training that induce better understanding of training dynamics, and bring about practical model improvements.

## 6.1   Related work

We note additional connections to existing literature here. The common understanding is that learning in networks is sparse; a subnetwork [6], or a random subspace of parameters [19] is sufficient for optimization and generalization. Our method provides an additional, more accurate, measure of usefulness to characterize per-parameter contribution. A similar work [34] defines per-parameter importance in the same vein but is computed locally with the mini-batch gradient, which overestimates the true per-parameter contribution to the decrease of loss of the whole training set.

Several previous works have increased our understanding of the training process. Alain and Bengio [2] measured and tracked over time the ability to linearly predict the final class output given intermediate layers representations. Raghu et al. [22] found that networks converge to final *representations* from the bottom up, and class-specific information in networks is formed at various places. Shwartz-Ziv and Tishby [25] visualized the training process through the information plane, where two phases are identified as empirical error minimization of each followed by a slow representation compression. There measurements are developed but none have examine the process each individual parameter undergoes.

Methods like saliency maps [27], DeepVis [33], and others allow interpretation of representations or loss surfaces. But these works only approach the end result of the model, not the training process in progress. LCA can be seen as a new tool that specializes on the microscopic level of details, and such inspection follows through the whole training process to reveal interesting facts about learning. Some of our findings resonate with and complement other work. For example, in [35] it is also observed that layers have heterogeneous characteristics; in that work layers are denoted as either "robust" or "critical", and robust layers can even be reset to their initial value with no negative consequence.

## 6.2   Future work

There are many potential directions to expand this work. Due to the expensive computation and the amount of analyses, we have only tested vision classification tasks on relatively small datasets so far. In the future we would like to run this on larger datasets and tasks beyond supervised learning, since the LCA method directly works on any parameterized model. An avenue to get past the expensive computation is to analyze how well this method can be approximated with gradients of loss of a subset of the training set. We are interested to see if the observations we made hold beyond the vision task and the range of hyperparameters used.

Since per-weight LCA can be seen as a measurement of weight importance, an simple extension is to perform weight pruning with it, as done in [6, 36] (where weight's final value is used as an importance measure). Further, if there are strong correlations between underperforming hyperparameters and patterns of LCA, this may help in architecture search or identifying better hyperparameters.

We are also already able to identify which layers or parameters overfit by comparing their LCA on the training set and LCA on the validation or test set, which motions towards future work on targeted regularization. Finally, the observations about the noise, oscillations, and phase delays can potentially lead to improved optimization methods.

# Acknowledgements

We would like to acknowledge Joel Lehman, Richard Murray, and members of the Deep Collective research group at Uber AI for conversations, ideas, and feedback on experiments.

## Footnotes

[1]We rarely see "neither", or zero-impact parameters in CIFAR networks, but it can be of a noticable amount for MNIST (around 20% for MNIST–FC; see Figure 3), mostly due to the many dead pixels in MNIST.

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
