[Supplementary Material · only_si_1029.pdf]

# Supplementary Information for:
# LCA: Loss Change Allocation for
# Neural Network Training

## S1   Supplementary results: Method

Table S1: Summary of errors of the LCA method with the Runge-Kutta method (RK4) used in our analyses, as well as the first order Taylor approximation (FO) for comparison. "Total error" is the percent error of total change in loss based on LCA vs. actual total change in loss over all iterations, where negative means that LCA gave a lower final loss. "Average iteration error" is the absolute error in one iteration, averaged over all iterations. Positive and negative errors at individual iterations could hypothetically cancel out somewhat when summed over all of training, though this is clearly not the case for first order, as it consistently and severely overestimates how much the loss decreases. This conforms with observations of the loss landscape being biased toward positive curvature. Reported numbers are averaged over 3 runs per configuration.

|  | Total error, RK4 | Total error, FO | Average iteration error, RK4 | Average iteration error, FO |
|---|---|---|---|---|
| MNIST-FC, SGD mom=0 | -0.27% | -4249.64% | 6.11E-05 | 0.11400 |
| MNIST-FC, SGD | 1.08E-03% | -171.10% | 2.94E-06 | 0.00459 |
| MNIST-FC, Adam | 3.64E-03% | -75.31% | 2.31E-06 | 0.00199 |
| MNIST-LeNet, SGD | 1.87E-02% | -274.84% | 3.45E-06 | 0.00838 |
| MNIST-LeNet, Adam | 0.0262% | -498.67% | 2.30E-06 | 0.01499 |
| CIFAR-ResNet, SGD | 0.0345% | -1189.71% | 1.37E-05 | 0.01018 |
| CIFAR-ResNet, Adam | 0.0537% | -923.88% | 1.07E-05 | 0.00793 |
| CIFAR-AllCNN, SGD | 0.1374% | -1371.69% | 4.51E-06 | 0.00662 |
| CIFAR-AllCNN, Adam | 0.0745% | -972.61% | 2.77E-06 | 0.00482 |

## S2   Supplementary results: Direct visualization

Examples of additional direct visualization methods shown in Figure S1 and Figure S2. Section S6 shows other possible visualizations with aggregations over neurons or channels.

Figure S1:  First layer of MNIST–LeNet with Adam. Rather than displaying all parameters in one iteration, we can sum up parameters within each output channel of this layer. The other axis can now be used to display all iterations.

Figure S2: Cumulative LCA for individual parameters in the first layer of MNIST–LeNet with Adam. Left: the 50 (out of 500) most helpful parameters (most negative LCA). Right: a random set of 50 parameters. You can see that the typical parameter's cumulative LCA drops quickly near the beginning and then continues to wiggle slightly after flattening out. Some parameters are mostly flat after the initial drop, but others continue learning slightly until the end.

## S3   Supplementary results: Learning is very noisy

We provide plots from Section 3 for all networks here in Figure S3, Figure S4, Figure S5, and Figure S6.

Plots of oscillation shown in Figure S7 and Figure S8 for ResNet, and additional oscillation measurements in Table S2 for all networks.

Adjusting hyperparameters has some effect on the percent of parameters helping, shown in Figure S9. However, the percent helped remains within a small range, especially when ignoring points of significantly worse test performance (rightmost points for momentum and mini-batch size).

Figure S3: MNIST–FC. Top: SGD with no momentum, middle: SGD with momentum = 0.9, bottom: Adam. Figures display the same measurements as in Figure 3, except the histogram of all LCA values now shows all values rather than ignoring the 1% tails.

Figure S4: MNIST–LeNet. Top: SGD, bottom: Adam

Figure S5: CIFAR–ResNet. Top: SGD, bottom: Adam

Figure S6: CIFAR–AllCNN. Top: SGD, bottom: Adam

Figure S7: Oscillations for CIFAR–ResNet with SGD. Left: number of times the weight movement oscillates. Right: number of times the gradient crosses zero from one iteration to the next. Values shown are number of iterations out of 5000, averaged over all parameters within a layer. The averages over the entire network are 741.9 for weight turns and 525.8 for gradients crossing zero. Note that the first and last layers oscillate more than their neighboring layers, which is interesting given that those layers hurt (Section 4), but this is only a correlation as oscillations do not explain why something would bias towards helping or hurting.

Table S2: Two metrics on oscillation: for each parameter, we look at the weight movement and count how often it switches direction (derivative of weight value changes sign) from one iteration to the next. We also look at the gradient for that parameter, and count how often it crosses zero (changes sign) from one iteration to the next. We convert both these into average frequencies over the training process, and then average those over all parameters in the network. These two measures are related – if a parameter oscillates around a local minima, its gradient would cross zero every time the weight changes direction – but due to noise, they do not have to correspond 1:1.

| Network | Number of iterations per weight movement direction change | Number of iterations every time gradient crosses zero |
|---|---|---|
| MNIST–FC, SGD mom=0 | 3.49 | 3.48 |
| MNIST–FC, SGD | 13.57 | 11.68 |
| MNIST–FC, Adam | 12.68 | 12.71 |
| MNIST–LeNet, SGD | 10.29 | 9.37 |
| MNIST–LeNet, Adam | 16.86 | 15.37 |
| CIFAR–ResNet, SGD | 6.74 | 9.51 |
| CIFAR–ResNet, Adam | 6.76 | 9.81 |
| CIFAR–AllCNN, SGD | 7.06 | 11.18 |
| CIFAR–AllCNN, Adam | 6.76 | 10.37 |

Figure S8: Oscillations of two individual parameters of the last kernel layer in CIFAR–ResNet with SGD. We look at iterations 50-200 (first 50 skipped due to large fluctuations), and display the parameter that hurt the most (top) and the parameter that helped the most (bottom) in these given iterations (net LCA of +3.41e-3 and -3.03e-3, respectively). Here, we visualize the weight movements (orange) along with their gradient values (blue) of the loss of the whole training set w.r.t. that parameter. Note that the trajectory of gradients and weights have similar shapes, indicating that these parameters are oscillating back and forth over a parabolic local minima that is shifting slightly (as the other parameters of the network are changing). During one back-and-forth cycle, the parameter will help, then hurt once the gradient crosses zero but momentum causes the weight to keep moving in the same direction, then help as the weight movement switches direction, and then hurt again when the gradient crosses zero again. This swinging LCA is depicted in the green and red bars. Because of these oscillations, the parameters end up helping approximately half the time and hurting the other half. While this behavior does not account for all the noise in other parameters and other iterations, it is commonly present.

Figure S9: Effects of different hyperparameter values on the percent of parameters helped per iteration, for CIFAR–ResNet with SGD. Left: learning rate, middle: momentum, right: mini-batch size. While there is some change in the percent helped, it does not go below 50.3% or above 51.6% within hyperparmeter ranges that still allow the network to learn. This excludes the configuration of momentum = 0.99 with 53.5% helped, where train and test performance have both degraded, and the number of parameters with zero LCA has actually significantly increased (resulting in 27.6% helped, 48.5% zero, 23.9% hurt). Also, as we increase the mini-batch size to 5000 (1/10th of the training data), the percent helped reaches 56.2%, but this is not practical as the test performance has become significantly worse.

(a) Actual distribution vs. Gaussian

(b) Cumulative LCA, actual

(c) Cumulative LCA, Gaussian simulated

Figure S10: A depiction of how LCA distributions are significantly more heavy-tailed than a Gaussian distribution. A kurtosis test on the actual distribution against a Gaussian gives excess kurtosis of 10420 and a p-value of effectively 0. The actual distribution has kurtosis of 2141, averaged across iteration intervals

# S4 Supplementary results: Some layers hurt overall

Additional plots: Figure S11 comparing 3 different networks, Figure S12 demonstrating the phenomenom of layers hurting for additional network architectures, Figure S13 for experiments concerning the first layer of CIFAR–ResNet, SGD, and Figure S14 concerning the last layer of CIFAR–AllCNN, SGD.

Figure S11: LCA summed over all of training, across each layer. Bias and batch norm layers are combined into their corresponding kernel layers. Left: MNIST–FC, middle: MNIST–LeNet, right: CIFAR–ResNet, all using SGD. While there is variation in the FC and LeNet layers (magnitudes are somewhat correlated to the size of the layer), they all are helping with negative LCA. On the other hand, the first and last layers of the ResNet strangely have positive LCA.

Figure S12: LCA of layers for two VGG-like models to further demonstrate that layers hurting is not a special one-off observation. It is also interesting to note that multiple layers hurt in these networks. Left: VGG-5 has 3 conv2D layers with 3x3 filters and {64, 128, 256} output channels, each followed by max pooling of stride 2, and then two fully connected layers with {512, 10} output units. Right: VGG-9 is the same as the VGG-11 used in [26] except with the last 2 conv2D layers removed, half the output channels in the remaining conv2D layers, and {512, 512, 10} output channels on the fully connected layers.

Figure S13: Left: LCA summed over all of training and across each layer of CIFAR–ResNet on SGD. Bias and batch norm layers are combined into their corresponding kernel layers. Blue represents a normal run configuration, and other colors show various experiments on the first layer. When the first layer uses a 10x smaller learning rate than the other layers (orange), per-layer LCA does not change much. While the "first layer frozen" runs (green) no longer hurt in the first layer (since the layer parameters are frozen from the beginning), the other layers, especially the next two, do not help as much. A similar effect is seen when we freeze the first layer at its LCA argmin (red); while we force the first layer to have negative LCA, the others have slightly more positive LCA, thus cancelling out any improvements. Middle: resulting train loss for each run configuration and standard deviations. Right: a typical cumulative trajectory of the first layer's learning, which helps in the first few hundred iterations and then increasingly hurts. The "freeze first layer at argmin" lets the layer help first before freezing it, but that still doesn't improve performance.

Figure S14: LCA for AllCNN layers. Last layer hurts in a regular run (blue), but freezing the last layer at initialization (orange) results in a worse overall loss (shown on the right).

# S5   Supplementary results: Learning is synchronized across layers

Figure S15: Peak learning iterations by layer by class on CIFAR–ResNet. We consider the first 400 training iterations, by which point the network achieves a test accuracy of 65%. We plot the top 20 iterations by LCA for each class and each layer, where that iteration represents a local minimum for LCA. The layers are ordered from bottom to top. Points highlighted in orange represent iterations where 25% to 50% of the kernel layers (6 to 10) had peak learning for that particular class, and there are 16.6 lines on average. Points highlighted in red represent iterations where at least 50% of the kernel layers (11 or more) had peak learning for that particular class, and there are 5.8 lines on average.

Figure S16: Peak effective gradient iterations by layer by class on MNIST–FC.

Figure S17: Peak learning iterations by layer on MNIST–FC. Eeah row represents a layer in the 3-layer FC network, orderred from bottom to top. Dots indicate top 20 moments of learning. and marked in red whenever synchronized across all layers. In this example 12 out of 20 moments are synchronized. The number of synchronized learning iterations is significantly more than chance, with a p-value of <0.001.

Figure S18: Peak effective gradients iterations by layer on MNIST–FC.

Figure S19: Peak weight movement iterations by class on MNIST–FC.

# S6    Supplementary results: Additional observations

## S6.1    Trajectory of parameters and temporal correlations

Our method allows for each individual parameter to have its own "loss curve", or, cumulative LCA, as seen in Figure S2. Interestingly, parameters going into the same neuron tend to have similar loss curves, meaning that they learn together. This can also be seen in the animations in Figure 2. We prove this concept with a correlation metric and statistical test. We conduct experiments for every layer in a network (focusing on kernel parameters, or weights, and ignoring bias and batch normalization parameters), calculate correlation coefficients of pairs of parameters, and apply Kolmogorov-Smirnov test to measurements for statistical significance. We find significantly stronger correlations between parameters of the same inputs or outputs than a random baseline, as depicted in Figure S20.

## S6.2    Class specialization in neurons

It is generally known that earlier layers in a neuron network learn more general concepts than later layers. This is akin to measuring the degree to which individual neurons specialize in learning specific classes. We can now show precisely how specialized neurons are and how this pattern evolves as we go deeper in the network.

We can also identify neurons that specialize in certain classes and visualize their behavior. For example, we look at two neurons in Figure S22 that concentrated on learning one and two classes, respectively. A saliency map using amount helped for each neuron gives us some insight into what that neuron is learning. A similar plot using just the weight values do not hold a clear pattern.

Figure S20: Correlation of weights within inputs and outputs, for every kernel layer of CIFAR-ResNet. **(Left)** Schematic indicating how weights belonging to the same input/output is like. **(Right)** Measured correlations for each layer. Multiple lines indicate multiple runs. For each layer, for each input/output, take all the weights going belonging to it, calculate pairwise correlation coefficients and the average of them. Then average through all nodes of that layer. Baseline for it is a constructed "fake node" with the same number of weights (or the most that exist), where no pair is from the same input or output.

Figure S21: **(Left)** MNIST–FC. **(Right)** CIFAR–ResNet. Fraction of neurons that concentrate on learning 1, 2, or 3 classes. For each neuron in each layer, we compute the ratio between amount helped for the top 1, 2, or 3 class(es) and the total amount helped for all positively benefited classes. We then find the fraction of neurons in each layer where the top 1, 2, or 3 class(es) contributed more than 80% of total learning.

Figure S22: **(Top)** An example of a first layer neuron that concentrated on learning the 0 class. **(Bottom)** An example of a first layer neuron that concentrated on learning the 0 and 8 classes. **(Left)** Cumulative amount helped by class. **(Right)** Plot of the 784 parameters within this neuron reshaped as a 2D image, colored by the LCA for each parameter.

# S7 Additional details on model architectures and training hyperparameters

All layers in network are followed by ReLu nonlinearity, and weights are initialized according to the He-normal distribution [10]. Exact implementation details can be found in our public codebase at `https://github.com/uber-research/loss-change-allocation`.

**MNIST–FC** We use a three-layer fully connected network, of sizes 100, 50, 10. For simplicity, no batch normalization or dropout was used.

**MNIST–LeNet** First conv layer is 5x5, 20 filters, followed by a 2x2 max pool of stride 2. Second conv layer is also 5x5, 40 filters, followed by a 2x2 max pool of stride 2 and dropout of 0.25. The result is flattened and fed through two fully connected networks with output sizes 400 and 10.

**CIFAR–AllCNN** AllCNN is the same as described in [29], with 9 convolutional layers followed by global average pooling. Batch normalization [13] and dropout [11] are used. The table below lists the size of layers and where batch normalization and dropout are added.

| | |
|---|---|
| 3x3 conv, 96 | followed by batch normalization |
| 3x3 conv, 96 | followed by batch normalization |
| 3x3 conv, 96 | Stride 2, followed by 0.5 dropout |
| 3x3 conv, 192 | followed by batch normalization |
| 3x3 conv, 192 | followed by batch normalization |
| 3x3 conv, 192 | Stride 2, followed by 0.5 dropout |
| 3x3 conv, 192 | followed by batch normalization |
| 1x1 conv, 192 | followed by batch normalization |
| 1x1 conv, 10 | |
| Global average pooling | |

**CIFAR–ResNet20** Each residual block consists of two 3x3 conv layers with the specified number of filters. Shortcuts are added directly if the number of filters is the same between blocks, otherwise the dimension change is done by a 1x1 conv with stride 2.

| |
|---|
| 3x3 conv, 16 |
| [Residual block of 16] x 3 |
| [Residual block of 32] x 3 |
| [Residual block of 64] x 3 |
| Global average pooling |
| Fully connected, 10 outputs |

**Hyperparameter search** We adjusted learning rate based on validation accuracy. The range we tried is an approximate log scale [1, 2, 5] times different powers of ten from 0.0005 to 1. Early stopping iteration was selected by when validation accuracy has flattened and train is mostly complete. For momentum, batch size, and dropout rates, we used reasonable and common values. We did not tune these (unless noted in special experiments) as they worked well and the exact values were not important. We used the given training/validation/testing split in the MNIST and CIFAR datasets.

For the LCA method, we also tried trapezoid rule, midpoint rule, and Boole's rule, and found that the Runge-Kutta method worked best given the same amount of computation.

**Learning rates used** The following learning rates are used in default experiments (SGD uses 0.9 momentum if not otherwise stated):

| | SGD | Adam |
|---|---|---|
| MNIST–FC, no momentum | 0.5 | N/A |
| MNIST–FC | 0.05 | 0.002 |
| MNIST–LeNet | 0.02 | 0.002 |
| CIFAR–ResNet | 0.1 | 0.005 |
| CIFAR–AllCNN | 0.1 | 0.001 |

## S8  Computational considerations

Consider the two terms of Equation 3. The second term, $\theta_{t+1}^{(i)} - \theta_t^{(i)}$, depends on the path taken by the optimizer through $\theta$ space, which in turn depends only on the gradients of mini-batches from the training set. This term is readily available during typical training scenarios without requiring extra computation. In contrast, the first term, $\nabla_\theta L(\theta_t)$, is computed over the entire training set and is not available in typical training. It must be computed separately, and because it requires evaluation of the entire training set, this computation is expensive (if the entire training set is $N$ times larger than your mini-batch, each iteration would take approximately $N$ times as long). Evaluating the loss and gradients over the entire training set at every training iteration may seem intractably slow, but in fact for smaller models and datasets, using modern GPUs we can compute this gradient in a reasonable amount of time, for example, 0.2 seconds per gradient calculation for a simple fully connected (FC) network on the MNIST dataset or 9 seconds for the ResNet-20 [9] model on the CIFAR-10 dataset. These times are quoted using a single GPU, but to speed calculations we distributed gradient calculations across four GPUs (we used NVIDIA GeForce GTX 1080 Ti). Thus, although the approach is slow, it is tractable for small to medium models.

| Model | Number of trainable params | Training time and iterations used | Time per iteration of gradient calculations on one GPU | Storage space per iteration |
|---|---|---|---|---|
| MNIST–FC | 84,060 | 4 min, 880 iterations | 0.4 s | 300 kb |
| MNIST–LeNet | 808,970 | 11-12 min 880 iterations | 2.4 s | 3.5 mb |
| CIFAR–ResNet20 | 273,066 | 120-125 min, 5000 iterations | 18-20 s | 1.2 mb |
| CIFAR–AllCNN | 1,371,658 | 270-280 min, 5000 iterations | 36-42 s | 6 mb |