[Reviews · NeurIPS 2019]

Reviewer 1



Originality: While lots of works have studied the property of the endpoint found by SGDs, the literature looking at the SGD training dynamics in the context of deep neural networks is sparser, and the loss contribution metric appears novel to me. The paper is therefore original from that aspect. Quality: The paper is in general of good quality. However, few specific points could be improved: - It would be nice to characterize the approximation errors introduced by the first order taylor expension - Authors claim that the Loss contribution is grounded while other Fisher information-based metrics heavily depends on the parametrization chosen. Could the authors expend on this point and provided a more detailed comparison between LC and the metrics introduced in [1] and [13] - In the introduction, authors claim that entire layers drift on the wrong direction during training. However, in the section, this observation only seems to apply to CIFAR/MNIST Resnet. It would be nice to characterize how robust is this observation. Clarity: The paper is clear and enjoyable to read. Significance. Looking at the training dynamics is an important topic. The authors propose a new metric to study the dynamics and provide nice empirical observation (learning is noisy, some layers sometime drift in wrong direction, learning is synchronized between layers). However, it is not clear how significant the loss contribution metric is, my main concerns are: - How does the LC compare with previously define metrics [1,13]. In which case the LC is more informative? - Is LC informative of the quality of the final point found by the optimizers? LC performs the dot products between the update and the batch gradient. It is not clear to me why using the batch gradient is a sensible thing to do. In particular, would networks training using batch-gradient descent using higher LC? We know that in practice, large-batch gradient find solutions that exhibit worse generalization - Would it make sense to use the gradient computed on the validation set to have a better estimate of the expected loss gradient instead of the empirical one? Update: Thank you for your rebuttal that did a convincing job about the utility of their metrics wrt to the FIM. On the other hand, after discussion, I also agree with R2 that further empirical validation is required to ensure the the metric can find neuron that are "hurting/helping" in training, or more generally why LC is informative of the quality of the final point found by the optimizers. In light of the second review, I find that the paper is borderline and I decided to keep my score. I do think that the idea explored in this paper is an interesting one and I encourage the authors to continue working in that direction.

Reviewer 2



Efforts to shine light on the black-box optimization process of large over-parameterized networks is an interesting and challenging topic. The authors investigate this problem by analyzing the per-parameter contribution to any loss function reduction. The paper is well-written overall and makes a few new and interesting observations. There are however, some major concerns I have with the paper that I detail next. - I think that the "help" or "hurt" heuristic is too simplistic and myopic. The authors claim that if the contribution of a parameter at any given iteration is negative, then the parameter is said to have hurt the loss reduction. While this is true in the mathematical sense, I feel that this is too simplistic. A local increase in loss function may lead to an eventual (greater) decrease contribution of the same parameter. By only looking at individual iteration snapshots, the authors have no way of accounting for such an effect. If the authors instead choose, say, an RMS value of the contribution, the interpretation of the term will be less myopic. Along the same lines, it might also be interesting to investigate the nodes that are maximally hurtful/helpful. Problems such as dead ReLUs (stemming from poor initialization) or connection to infrequent features might disable neurons (and hence, paths) and it would be interesting to see if the author's approach could discover these. - As a thought experiment to verify the above claim, I suggest the following experiment. At any given iteration, take a learning step and compute the sets of helpful, hurtful and indifferent parameters. Then, undo this iteration and take the same step but only for the variables which were helped. If second-order effects are not present and this myopic view is true, then such an experiment should yield similar outcomes to the original training trajectory (or better). - The approach ignores curvature from the discussion. Rather that using simply the first-order terms, maybe the authors could try using higher-order terms too? This may be expensive in some circumstances but an approximation of this RMS values of the moments of the gradients (similar to what Adam/Adagrad maintain) might be worthwhile too. - "Freezing the last layer results in significant improvement." is a known phenomenon, see "FIX YOUR CLASSIFIER: THE MARGINAL VALUE OF TRAINING THE LAST WEIGHT LAYER" from ICLR 2018. - The paper is missing actionable uses from the analysis. While a detailed analysis is enough for publication to NeuRIPS, I feel that this paper is incomplete without some sample uses of the analysis. The authors discuss some possibilities in the last section (using LC for identifying over-fitting, or for architecture search) but stay short of presenting them. I highly encourage the authors to have exemplar uses of their analysis in the paper. -- UPDATE -- I read the rebuttal and other reviews and I increase my score from 4 to a 5. I feel that, especially in the light of their rebuttal, a score of 4 is unfair to the authors. I wish to note that the author's rebuttal was very well presented. However, I am still concerned about the metric. While it has benefits over FIM, I feel that there is inadequate validation of whether this metric highlights what is intended. The authors do not discuss the experiment I suggested in my review to tease this effect. Further, the claim about myopic vs aggregate effects (Rebuttal #5) is not convincing. If the myopic and aggregate views are different, that requires a reconciliation with some of the other claims.

Reviewer 3



The main contribution is the loss contribution metric. The metric is then applied to analysing deep neural networks. It is a challenging task to define a clear and interpretable metric that shows a new suprising perspective on the training of deep networks and the authors managed to do it. I believe that the experiments clearly demonstrate utility of the metric and the results are surprising. The paper is very well written. I was a bit let down that there is no novel practical tricks presented (see below for details). Nevertheless, the paper will be clearly of interest to the community, and I am quite optimistic that future work will bring more practical applications of the developed metric. Detailed comments 1. Showing oscillatory-like behavior in training is not very novel. "Walk with SGD" (https://arxiv.org/abs/1802.08770) and "On the Relation Between the Sharpest Directions of DNN Loss and the SGD Step Length" (https://arxiv.org/abs/1807.05031) seem to already show a quite related dynamic in training. The first paper shows that the gradient oscillates (i.e. cosine is negative between subsequent iterations). The latter paper shows that there are directions in the weight space (corresponding to the largest eigenvalues of the Hessian) where training is unstable. What is novel, I agree, is that such a behavior happens on the parameter-level, that it is as dominant as shown, and how parameters switch between helping and hurting. It would be nice to contextualize prior work in a bit better. 2. Instability of the last layer was discussed by some prior work, e.g. https://openreview.net/forum?id=r14EOsCqKX. Freezing layers, especially the last one, is also not novel. In https://openreview.net/forum?id=r14EOsCqKX they also freeze the last layer. 3. The paragraph "Learning is heavy-tailed" could be made a bit more precise. For instance, how would refining the view of learning as a Wiener process alter the conclusions made by these papers? It wasn't very clear to me. 4. I would, though it is personal taste, remove exclamation marks. I think using them is not the best practice in scientific writing. 5. Experiments show on the example of the first layer that freezing a layer that hurts might hurt even more because other layers help less. This is not very intuitive. It also seems to limit applicability of the developed metric (if the metric shows layer hurts, we do not know if we should improve it or not). If possible, it would be nice to explain the result better. Update Thank you for the well-written rebuttal! I decided to keep my score. I would encourage authors, in case paper ends up rejected, to run experiments suggested by one of the reviewers and examine effect of skipping updates that are calculated to be negative, or any related experiment that would pinpoint the causal effect these dynamics have on the training performance.

[Author Response · NeurIPS 2019]

We thank the reviewers for their critical feedback. Quite honestly there were large gaps in the explanation and analysis (as loosely clustered below: 10 of them), and we've made serious improvements to address nearly all:

**1. Missing actionable use cases (R2, R3); implications of "50.7% helped" unclear (R3)**. Our paper aims to increase the understanding of neural networks in the same vein as insightful but not directly actionable papers like "Intriguing properties of Neural Networks" (Christian Szegedy et al, 2013) and "Opening the Black Box..." (Shwartz-Ziv and Tishby, 2017). We did find one use case (identify layers for freezing), but the primary goal of our paper is to provide a new measurement device (LC) and to clarify and update our mental models of NN training by using it.

**2. Why is LC a good metric? Compare with FIM from [1,13] (R1)**. We believe both are useful. LC answers the specific question of "how do we allocate change in loss?" and is beneficial vs FIM because it is *grounded to the loss* and *signed*. *Grounded*: While [1] and [13] are illuminating, the FIM metric used may be rescaled arbitrarily (e.g. multiply one relu layer weights by 2 and next by 0.5: loss stays the same but FIM of each layer changes and total FIM changes). In contrast, LC is grounded, so its units are the same as loss (e.g. bits or nats for cross entropy) and its scale is fixed. *Signed*: FIM is unsigned and thus could not yield our "50.7% help" nor "some layers move backward" conclusion. We have clarified these points in the paper. *[See Update A in the figure below]*

**3. Justify use of batch gradient (R1)**. This part was poorly presented; we have updated the text *[Update B]*. To measure training instead of training confounded with issues of memorization vs. generalization, we use the *train* gradients instead of *val*. Observations like the last layer hurting are also more surprising on train vs. val. We use *full-batch* gradients for analysis instead of single mini-batch gradient to measure learning in as noise-free a way as possible. Note that the optimizer only ever sees mini-batch gradients, as is the usual case for SGD or Adam.

**4. Missing higher order terms, ignores curvature (R2); characterize approximation errors from first order (R1)**. This part was confusingly presented; text updated to clarify *[Update C]*. Curvature is not ignored, as LC is approximated with RK4, which is fourth-order accurate. First order was only mentioned as an example to illustrate the concept. Approximation errors on all networks with both first order and RK4 have been added *[Update D]* (good idea!). In short, first order can be off by a lot (it always overestimates the decrease in loss), but RK4 is within 0.3% total loss change.

**5. "Help" and "hurt" are too simplistic and myopic, loss may locally increase then decrease later (R2)**. Each LC measure concerns the impact on loss at each specific iteration, and we agree that periods of hurting may be followed by helping and vice versa. We see this often (e.g. see Fig 3d). By summing LC over time, we can take arbitrarily less mypoic views (e.g. see Figs 4 and 5). We believe both microscopic and aggregate views can be useful.

**6. Last layer: different behavior and benefit of freezing already known (R2, R3)**. Thanks for these great references; we have added a discussion *[Update E]*. We build on this previous work by proposing a theoretical explanation (phase-delay) and validating the theory with careful experiments. See also: new multi-layer results in #7.

**7. How universally does last layer hurt? (R1)**. We don't claim this to be a universal phenomenon, but when it occurs LC allows us to uncover it and take beneficial actions such as freezing layers. It is also not just a one-off observation: in addition to ResNet, it also happens for AllCNN (mentioned in paper) where the last layer is conv, and we have now measured it in VGG-based models as well, where the last 2-3 FC layers show the phenomenon *[Update G]*.

**8. Citations, oscillations not novel (R3)**. Thanks for these great references; we had not meant to claim discovery of oscillation and have updated the text to clarify *[Update F]*. LC merely tracks such behavior on a per-parameter level and can expose, for example, whether a parameter or layer is slowly making progress or the reverse over time.

**9. Intuition of results of freezing the first layer (R3)**. Indeed, given the observed behavior of freezing the first layer (that other layers then help less), we should be careful not to over-interpret a positive per-layer LC as directly indicating that that layer should be removed or frozen. The interaction effects between layers would seem to matter a lot, though we do not fully understand these effects yet (though our phase-delay results are a step in the right direction).

**10. Learning is heavy tailed section not precise (R3)**. Thanks for the suggestion. We've updated it to a more precise characterization: the LC distribution is closely modeled by a heavy-tailed Weibull distribution *[Update H, dashed line shows the Weibull distribution, parameter $k = 0.53$ in example]*. This refines a detail in our mental model of training.

A, B, C. Clearer explanation of LC method and comparison with other methods

D. Added table of approximation errors for RK4 (used in main findings) and first order

**Update Highlights**

E. Citations for last layer behavior

F. Citations for oscillations

G. The last FC layers of two new VGG-styled networks also hurt

H. LC distribution closely fits a Weibull distribution (dashed line)

[Meta-Review · NeurIPS 2019]

There is some disagreement about this paper among reviewers. There is a common appreciation for this line of study and specifically the new loss contribution (LC) metric proposed. As many things about the training process of DNNs remains "mysterious", developing new and better "lenses" through which we can look at the inner workings of a DNN can be of great value for the field. The criticism in the less enthusiastic reviews is largely around "more effort": comparison to other approaches, more experiments, clarifications and improvements, making it more actionable. One can also give that a positive spin: there is a lot of interesting follow-up work to be done here. I want to give the paper the benefit of the doubt, encourage the authors to work on a revision that addresses some of the holes spotted by the reviewers (the rebuttal is already a great step in that direction).